

# Microbial community composition and function in an urban waterway with combined sewer overflows before and after implementation of a stormwater storage pipe

Kazuaki Matsui[1] and Takeshi Miki[2]

[1] Department of Civil and Environmental Engineering, Kindai University, Higashiosaka, Japan
[2] Faculty of Advanced Science and Technology, Ryukoku University, Otsu, Shiga, Japan

## ABSTRACT

When the wastewater volume exceeds the sewer pipe capacity during extreme rainfall events, untreated sewage discharges directly into rivers as combined sewer overflow (CSO). To compare the impacts of CSOs and stormwater on urban waterways, we assessed physicochemical water quality, the 16S rRNA gene-based bacterial community structure, and EcoPlate-based microbial functions during rainfall periods in an urban waterway before and after a stormwater storage pipe was commissioned. A temporal variation analysis showed that CSOs have significant impacts on microbial function and bacterial community structure, while their contributions to physicochemical parameters, bacterial abundance, and chlorophyll a were not confirmed. Heat map analysis showed that the impact of CSO on the waterway bacterial community structure was temporal and the bacterial community composition in CSO is distinct from that in sewers. Hierarchical clustering analysis revealed that the waterway physicochemical water qualities, bacterial community composition, and microbial community function were distinguishable from the upper reach of the river, rather than between CSO and stormwater. Changes in the relative abundance of tetracycline resistance (*tet*) genes—especially *tet*(M)—were observed after CSOs but did not coincide with changes in the microbial community composition, suggesting that the parameters affecting the microbial community composition and relative abundance of *tet* genes differ. After pipe implementation, however, stormwater did not contribute to the abundance of *tet* genes in the waterway. These results indicate that CSO-induced acute microbial disturbances in the urban waterway were alleviated by the implementation of a stormwater storage pipe and will support the efficiency of storage pipe operation for waterway management in urban areas.

Corresponding author
Kazuaki Matsui,
kmatsui@civileng.kindai.ac.jp

## INTRODUCTION

Cities tend to develop near watersheds because water is essential for urban life. Indeed, 55.2% of the world's population lived in urban areas in 2018, and this is expected to increase to 68% by 2050 (*United Nations, 2019*). Population growth in urban areas occurs on <3% of the terrestrial surface, but 60% of global residential water use is attributed to cities (*Grimm et al., 2008*). The dense population of urban areas has a consistent impact on water quality (*Mallin, Johnson & Ensign, 2009*; *Kaushal et al., 2015*) and biological habitat (*Hatt et al., 2004*; *Drury, Rosi-Marshall & Kelly, 2013*; *Hamdhani, Eppehimer & Bogan, 2020*) in urban rivers and other waterways.

Contamination of stormwater and sewage are a leading cause of urban waterway fecal pollution (*McLellan, Fisher & Newton, 2015*). Model simulations predict that the major sources of fecal coliform loadings to European rivers are domestic sewage, followed by private treatment, urban surface runoff, and manure application (*Reder, Flörke & Alcamo, 2015*). Ecologically, wastewater pollution can impact stream food webs *via* a combination of energy limitation to consumers and extirpation of pollution-sensitive top predators (*Mor et al., 2022*). Impervious surfaces (*i.e.*, concrete and asphalt) act as stormwater pathways and collect urban-derived pollutants into waterways. Non-point discharge from leaky or failing sewer pipes collects human-derived pollutants into waterways (*Newton & McClary, 2019*). Historically, sewer pipes have been constructed to drain sanitary sewage and discharge the water from heavy rain to prevent flooding (*Tibbetts, 2005*). A combined sewer system that carries both sewage and stormwater requires only one set of sewer pipes. This has the advantage of reducing construction and maintenance costs compared to separate sewer systems. Therefore, numerous older cities still have combined sewer systems. However, these may cause sewage to overflow into waterways during extreme rainfall events.

Combined sewer overflows (CSOs) have multiple effects on the waterway environment, including oxygen depletion caused by biodegradation of organic matter (*Seidl, Servais & Mouchel, 1998*; *Maki et al., 2007*); reduction of primary production by turbidity (*Lee et al., 2002*); increased levels of chemicals and heavy metals (*Zgheib, Moilleron & Chebbo, 2012*; *Xu et al., 2018*) and pathogenic and fecal microbes (*Kim et al., 2009*; *Fong et al., 2010*; *Newton et al., 2013*); altered microbial composition (*Fisher et al., 2015*; *Chaudhary et al., 2018*); and increased detection of antibiotic resistance genes (ARGs) (*Eramo, Delos Reyes & Fahrenfeld, 2017*).

Nevertheless, river managers tend believe that high-flow rivers do not suffer the impact of CSOs due to their capacity to dilute contamination (*Soriano & Rubio, 2019*). However, few studies have examined how urban discharge, such as CSOs, can alter the microbial ecosystem in urban stagnant water (*Gosset, Ferro & Durrieu, 2016*).

Pollutant discharge is higher in the initial period of a CSO (*i.e.*, the first flush effect) than in later periods. *Bertrand-Krajewski, Chebbo & Saget (1998)* hypothesized that the first flush transports at least 80% of the total pollutant mass in the first 30% of the volume discharged during rainfall events. The effects of first flush-driven short-term CSOs—such as high levels of dissolved contaminants and microbes—affect both water quality and

microbial community composition in the receiving waterway (*Lee et al., 2002*; *Riechel et al., 2016*). As various antibiotics have been detected in sewers, CSOs can be a source of antibiotics in the aquatic environment (*Zhou et al., 2013*). The released antibiotics can alter microbial community structure and lead to the prevalence of bacterial resistance to antibiotics (*Grenni, Ancona & Barra Caracciolo, 2018*).

However, first flush transports may not explain the total impact on the microbial ecosystem in urban waterways. *Eramo, Delos Reyes & Fahrenfeld (2017)* noted that the maximum detection of ARGs and a marker gene for human fecal indicator organisms did not coincide with the first flush transport even though CSO was found to be a source of ARGs. In this context, other studies noted the significance of stormwater in contributing ARGs and microbial loads to urban streams (*Baral et al., 2018*; *Garner et al., 2017*). Furthermore, the CSO impacts on microbial parameters, such as composition and function, will differ in each urban waterway. Thus, comparative evaluations of microbial parameters between CSOs and stormwater periods will further our understanding of the impact of allochthonous water on urban waterways.

This study evaluated the physicochemical water quality, microbial community structure, and microbial functional diversity at an urban waterway before and after stormwater storage pipe operation. We hypothesized that the CSO first flush temporarily shifts the waterway microbial composition toward that of the sewer, and the changes in microbial composition alter the microbial function. To estimate the ecotoxicological risk, EcoPlate-based techniques were used to characterize such functional changes (*Miki et al., 2018*). The abundance of tetracycline resistance (*tet*) genes was quantified to estimate the occurrence of CSO- or stormwater-mediated release of ARGs in the waterway. A survey was also conducted in a connected urban river to determine the shift introduced by the semi-closed, stagnant water in the waterway. These comparative surveys allowed us to estimate the impact of CSO on the waterway ecosystem and evaluate the efficiency of storage pipe operation.

## MATERIALS AND METHODS

### Study site

This study was conducted along the Higashiyokobori-gawa waterway (H-waterway). This 3.0-km artificial waterway was excavated approximately 500 years ago and runs north to south through the highly populated urban area of Osaka City, Japan (Fig. 1A). The upper part of the waterway follows the course of an elevated expressway. Thus, the expressway covers the waterway and stormwater is discharged from the elevated expressway into waterways (Fig. 1B). Downstream of the H-waterway is the east to west Dotonbori-gawa waterway (D-waterway), which was also excavated approximately 500 years ago. Water gates are at the north entrance of the H-waterway and the west exit of the D-waterway. These waterways are described in detail elsewhere (*Matsui, Fumoto & Kawakami, 2019*).

The 2.75 million residents of Osaka, Japan, live within 225.32 km$^2$ (*Osaka-city, 2021*). The sewage system in this area was developed after the 1950s. Because the city is in a low-lying, frequently flooding area, >97% of the sewer system is a combined sewer (*Matsumoto & Takayanagi, 2005*). The overflow sewer runs into waterways during heavy

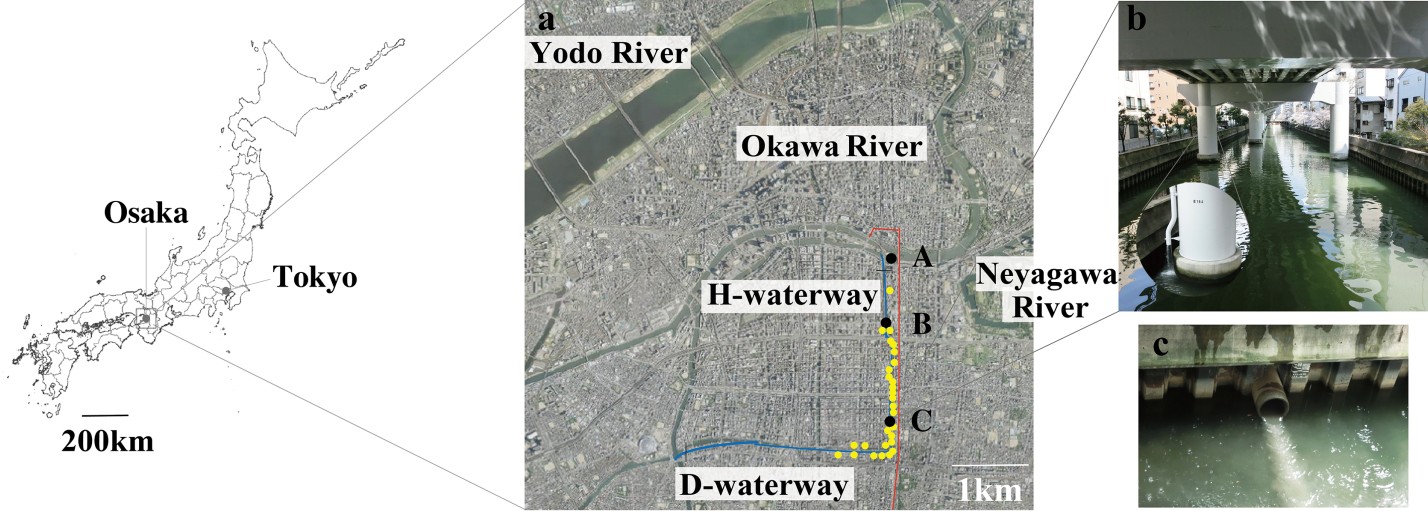

**Figure 1  Map of the study site.** (A) Sampling stations A to C (filled black dots) were chosen along with the Higashiyokobori-gawa waterway (H-waterway). Downstream of the H-waterway is the Doutonbori-gawa waterway (D-waterway), which flows from east to west. The water gates are shown as black bars. Release outlets (yellow-filled dots) and the underground stormwater storage pipe (red line) are also shown in the picture. The aerial photo was taken by the Geospatial Information Authority of Japan (GSI) in 2006. (B) The upper part of the H-waterway becomes the course of the expressway and entirely covered. The enlarged portion shows stormwater discharge from the elevated expressway into the waterway. (C) Combined sewer overflow into the waterway from a release outlet.           

precipitation (*Usuta, 2011*). There are 28 release outlets on the H-waterway and D-waterway; the CSOs discharge into waterways during around 70 out of 85 annual precipitation events (*Usuta, 2011*; *Matsui et al., 2017*). In a single CSO, release of 44–5,971 m$^3$ of a mixture of untreated sewage and stormwater runoff has been recorded from one discharge outlet on the H-waterway (*Yoshida, 2007*).

Water quality in the waterways was degraded by urbanization between 1950 and the 1970s. The water quality has been improved by construction of a sewer system and effluent control, dredging, and water replacement *via* water-gate management (*Nakatani, Nishida & Inoue, 2020*; *Usuta, 2011*). The annual mean biochemical oxygen demand in the waterways was ≥20 mg/L in 1970 but <4 mg/L in the 2000s. However, the waterways receive CSOs during heavy rainfall periods (Fig. 1C). To prevent CSO entry into the waterways, an underground stormwater storage pipe was constructed and commissioned by Osaka City in March 2015. The storage pipe (6-m diameter × 4.8-km length; 140,000-m$^3$ storage volume) was designed to capture the whole volume of stormwater runoff for the designed rainfall event, a 10-year return period rainfall event (60 mm/h) (*Matsumoto & Takayanagi, 2005*). The pipe was built below any service tunnels (*e.g.*, cables and gas) and subway lines, and was covered with 33–49 m of earth (*Usuta, 2011*).

The intake of the H-waterway is at the confluence of the Okawa and Neyagawa Rivers. The Okawa River is a diversion of the Yodo River, which flows from Lake Biwa. It has good water quality (annual mean total nitrogen and total phosphorus levels of 1.1 and 0.061 mg/L, respectively, in 2018) (*Osaka-Prefectural-Government, 2020*). The Neyagawa River, which experiences CSOs during heavy rain, runs east to west through a highly populated area and shows low water quality (annual mean total nitrogen and total

phosphorus levels of 6.3 and 0.29 mg/L, respectively, in 2018) (*Osaka-Prefectural-Government, 2020*). To maintain the water quality of the waterways, Osaka City operates the water gates daily to take in Okawa River water and block Neyagawa River water (*Usuta, 2011*).

## Sampling and physicochemical measurements

Three sampling stations were selected along the H-waterway (Fig. 1A). Station C is 2,500 m south of the water gate and surrounded by release outlets. Station B is 500 m south of the water gate, with two release outlets in close proximity. Station A is 100 m upstream of the water gate.

   Total precipitation amount, maximum intensity, and duration were obtained from the Japan Meteorological Agency (http://www.jma.go.jp/jma/). Based on the obtained information, rainfall events were categorized into three phases (Table S1). Strong (phases 1 and 3) and weak (phase 2) precipitation were distinguished based on an arbitrary value (total precipitation × maximum intensity ÷ duration) at a threshold of 10. Phase 3 was the period after the stormwater storage pipe had been commissioned.

   In each phase, we conducted four to six grab-water samplings (Table S2). Because the first flush discharge occurs in a very short time, the grab-water sampling allows us to adjust the timing with the first flush discharge from surrounding release outlets. The samples are a subset of the samples used for DNA analyses in our prior study (*Matsui et al., 2017*). On-site measurements of water temperature and pH were performed using a pH meter (D-71, Horiba, Kyoto, Japan). On-site measurements of conductivity and dissolved oxygen concentration were performed using a multiparameter meter (Multi 3420; WTW, Germany, Europe). Water samples were collected at 10–30 cm from the surface using a bucket with a rope from the shore (station A) or atop a bridge (stations B and C). A grab-water sewer inflow sample was obtained on January 13, 2016, at Tsumori wastewater treatment plant, where sewage was drifting down from areas around waterways.

   The collected water samples were kept in a cooler box containing ice packs and immediately transported to the laboratory. The temperature inside the cooler box was kept below 8 °C for sampling in January and March and was kept below 17 °C for sampling in May, July and September. All experimental treatments were conducted within 8 h of sampling. Soluble reactive phosphorus, nitrite, and nitrate concentrations were determined using an AACS-II (Bran+Luebbe Co., Germany, Europe) continuous flow system. Total phosphorus was converted to orthophosphate in accordance with the method established by *Menzel & Corwin (1965)*; it was then measured using an AACS-II system. Chlorophyll *a* was determined by a spectrophotometric method after extraction in *N,N'*-dimethylformamide (*Porra, Thompson & Kriedemann, 1989*). Bacteria were enumerated directly under an epifluorescence microscope by SYBR Green I staining (*Honjo et al., 2007*). Particulate organic carbon (POC) and nitrogen (PON) were trapped on a GF/F filter (GE Healthcare Life Science) and measured using a CHN micro-coder (JM10, J-SCIENCE LAB Co., Japan, East Asia). Dissolved organic carbon in GF/F filtrate was analyzed by a TOC analyzer (TOC-Vw; Shimadzu, Japan, East Asia).

## Analysis of microbial function

Culture-based microbial metabolic profiling was conducted using an EcoPlate (Biolog, Hayward, CA, USA), in accordance with the method established by *Miki et al. (2018)*. The EcoPlate quantifies the ability of a microbial community to utilize 31 distinct carbon substrates, by monitoring the color development of microplate wells during incubation. Briefly, 100 µL of water sample were directly inoculated into EcoPlate wells using a Repeater M4 Pipette (Eppendorf, Japan, East Asia). The plates were incubated at the sampling water temperature for 1 week in the dark. Color development was determined after 24, 48, 72, 120, and 168 h of incubation by assaying the optical density at 590 nm using a microplate reader (iMark; Bio-Rad, Hercules, CA, USA). The data were processed to assess the level of functional dissimilarity (*Miki et al., 2018*).

## DNA extraction

Microbial communities were collected from water samples by filtration of 100-mL subsamples onto a 0.2-µm nucleopore polycarbonate filter (47 mm diameter; Advantec, Japan). Each filter was stored at −25 °C until DNA extraction. For DNA extraction, a filter was transferred into a 2-mL beads tube from the ISOIL for Beads Beating kit (Nippon Gene, Japan, East Asia). DNA extraction was generally conducted in accordance with the manufacturer's protocol to maximize yield; an important modification was the use of chloroform extraction at the beginning of the process to solubilize the polycarbonate filter. The DNA recovery efficiency was 11.0–19.4% (Table S3).

## Analysis of bacterial community structure

We amplified and sequenced the V3–V4 region of the 16S rRNA gene (*Escherichia coli* positions 339 to 805) in accordance with the Illumina 16S metagenomic sequencing library protocol, using *Ex Taq* Hot-Start DNA polymerase (TaKaRa, Japan, East Asia). The cycle conditions were 95 °C (2 min); followed by 25 cycles of 95 °C (30 s), 55 °C (30 s), and 72 °C (30 s); and a final extension at 72 °C (5 min). Libraries were purified using AMPure XP (Beckman Coulter, Japan, East Asia). Illumina sequencing adapters from the Illumina Nextera XT Index Kit were added to the target amplicons in a second PCR step. The cycle conditions were 95 °C (2 min); followed by eight cycles of 95 °C (30 s), 55 °C (30 s), and 72 °C (30 s); and a final extension at 72 °C (5 min). Libraries were purified and quantified using the Qubit dsDNA HS Assay Kit (Thermo Fisher Scientific, Waltham, MA, USA). The purified amplicon libraries were combined in equal concentrations into a single tube. Sequencing of the combined amplicon library was conducted on the Illumina MiSeq at FASMAC (Japan). The obtained raw reads were trimmed with the Fastx toolkit v. 0.0.13.2 (*Gordon & Hannon, 2010*), filtered according to the Q20 score in sickle v. 1.33 (*Joshi & Fass, 2011*), and merged using FLASH v. 1.2.10 (*Magoč & Salzberg, 2011*). Chimeric sequences were verified using USEARH (*Edgar, 2010*) and sequences with 97% similarity were classified as an operational taxonomic unit. Taxonomy assignments were determined by the Greengenes classifier (*McDonald et al., 2012*). Sequence processing was conducted in QIIME v. 1.9.0 (*Caporaso et al., 2010*) using the default conditions.

## Screening for tetracycline-resistant bacteria

Culturable tetracycline-resistant bacteria were screened from sample waters taken at stations A, B, and C. Two hundred microliters of water were spread on MacConkey agar (Wako, Japan) containing 10 μg/mL tetracycline and incubated for 48 h at 30 °C. Colonies were streaked onto MacConkey agar containing 10 μg/mL tetracycline to obtain single colonies. Sixty-six colonies were screened by colony PCR (direct inoculation of the colony into a PCR tube) targeting eight tetracycline resistance genes (Table S4). The cycle conditions were 95 °C (1 min); followed by 30 cycles of 95 °C (1 min), annealing (1 min; temperatures shown in Table S4), and 72 °C (90 s); with a final extension at 72 °C (10 min). Cells from single colonies were cultured in Luria–Bertani liquid medium containing 10 μg/mL tetracycline (*Green & Sambrook, 2012*); total cellular DNA was extracted using the Wizard Genomic DNA Purification Kit (Promega). A 1.4-kb region of the 16S rRNA gene was amplified from total cellular DNA using two universal 16S rRNA gene primer pairs (Table S4) to identify bacterial taxa.

The PCR products were purified using AMPure XP (Beckman Coulter), ligated into pGEM-T Easy vector (Promega), and transformed into competent *E. coli* DH5α cells. The plasmid extracted from a single colony of each transformant was sequenced at the sequencing facility of Hokkaido System Science Co. (Sapporo, Japan) using BigDye Terminator v. 3.1 and an automated sequencer (Prism 3130; Applied Biosystems, USA). Sequences were compared with GenBank data using the Basic Local Alignment Search Tool (*Altschul et al., 1997*). We identified seven *tet*(A) isolates, one of which was cloned and sequenced.

## Real-time quantitative PCR

The primers and TaqMan hydrolysis probes used are listed in Table S5. Quantitative (q)PCR assays were performed using a Roche LightCycler 480 System (Roche Diagnostics, Germany, Europe). qPCR was performed in a 20-μL reaction containing 200 nM of each primer, 100 nM of each probe, and 1× LightCycler 480 Probe Master (Roche). Template DNA was added at final concentrations of 50 fg to 5 ng in a 5-μL volume. Plasmid clones of *tet*(A), *tet*(B), and *tet*(M) were obtained as described in "Screening for tetracycline-resistant bacteria" and used to construct standard curves (Fig. S1). The qPCR conditions were initial denaturation at 95 °C (10 min); followed by 45 cycles of denaturation at 95 °C (10 s), annealing at 60 °C (30 s), and extension at 72 °C (1 s). Each sample was analyzed in triplicate and crossing point (*Cp*; second derivative method) values were determined using the LightCycler 480 software version 1.51. Relative abundance was calculated by normalizing the *tet* copy number to the total number of bacteria.

## Data preprocessing and statistical analysis

The R program (v. 3.6.2; *R Core Team, 2020*) and the vegan package (v. 2.5.6; *Oksanen et al., 2008*) were used for data preprocessing and the statistical analysis.

*Data preprocessing*—For physicochemical parameters, soluble reactive phosphorus was excluded because data were missing for some dates; bacterial abundance and chlorophyll *a* were also excluded. We used the Euclidean distance to evaluate compositional dissimilarity

among samples. For microbial function (EcoPlate data), the integration method was used with multiple measurements during incubation with the maximum values of triplicate measurements on each plate (*Miki et al., 2018*). The Bray–Curtis distance was used to evaluate compositional dissimilarity. For bacterial community structure, although the frequency of singletons in the sequence data is important for estimating the coverage of sampling efforts, singletons are often included from sequence error (*Chiu & Chao, 2016*). We estimated the number of "true" global singletons and randomly deleted extra singleton sequences (*Chiu & Chao, 2016*). Next, based on the local sequence distribution including local singletons, the sampling coverage and rarefied local community sequences were estimated at a minimum coverage of 95.47%. We used the iNEXT package (v. 2.0.20) (*Hsieh, Ma & Chao, 2016*) to estimate the sampling coverage and used the resample() function for rarefaction in R. Standardized rarefied sequence data were used for the statistical analysis. To evaluate the compositional dissimilarity of the operational taxonomic unit dataset, the Hellinger distance was used.

*Statistical analysis*—The capscale() function in *vegan* package for principal coordinate analysis (PCoA) was used. We used permutation multivariate analysis of variance (PERMANOVA) in the adonis() function in *vegan* with 999 permutations to statistically test differences between groups for multivariate variables statistically (physicochemical characteristics, bacterial count, chlorophyll *a* level, microbial function, and microbial community structure) after checking the homogeneity of variance using PERMDISP and the betadisper() function in *vegan* with 999 permutations. We also used PERMDISP to evaluate the magnitude of temporal variations (Tables S6 and S7). For both PERMDISP and PERMANOVA, we first checked the overall significance >2 groups, but the pairwise comparison was conducted independently of its $P$ values. For hierarchical clustering, the hclust() function with the ward.D2 option was applied (*Murtagh & Legendre, 2014*). We used $P < 0.01$ as the significance level and performed Bonferroni adjustment for pairwise multiple comparisons (*i.e.*, $P < 0.01 \div 3 = 0.00333$ for comparisons of three groups). For graphics, ggplot2 was used.

## RESULTS AND DISCUSSION

### Effect of heavy rain on waterway water quality

We evaluated the effects of CSO on waterway water quality at station C (Table S9). A CSO-induced discharge of organic matter was frequently observed (*Servais, Seidl & Mouchel, 1999*). An increment of POC (from 1.1 to 2.9 mg/L) was observed durng the CSO in phase 1, while no DOC increment was observed. No increment of POC or DOC was observed in phases 2 and 3. The conducrivity (from 772 to 1,413 µS/cm) and PON (from 0.1 to 0.3 mg/L) also increased during CSO in phase 1. However, there increments were also observed in phase 3. Conductivity and PON may be less specific CSO indicators compared to organic matter. The low DO levels were mainly caused by bacterial oxygen consumption following the CSO (*Seidl, Servais & Mouchel, 1998*), while CSO did not affect DO levels in this study. Low DO levels were generally observed during the summer; some CSOs did not produce DO sags (*Irvine, McCorkhill & Caruso, 2005*). Because the water temperature in phase 1 was 7.0–7.4 °C, bacterial oxygen consumption was limited.

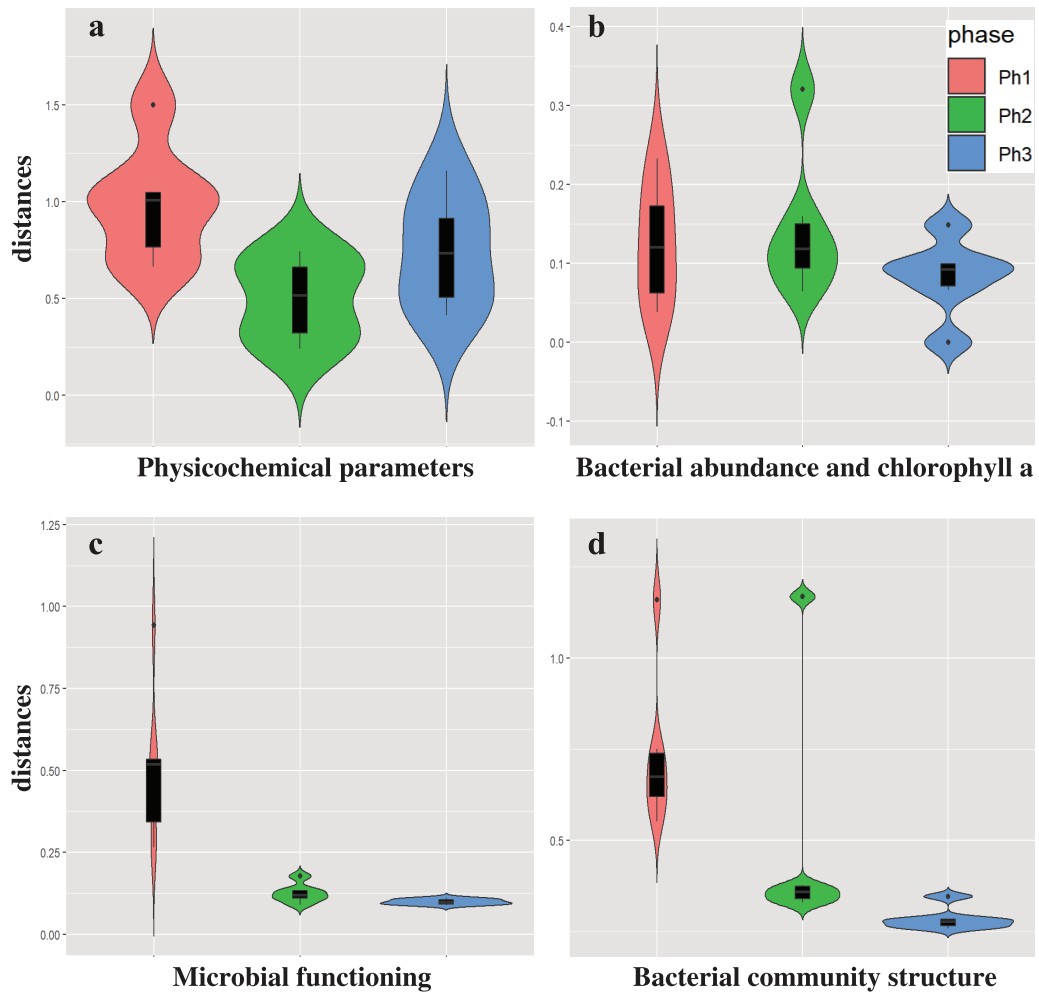

**Figure 2 The temporal variations of the CSO sensitive station C water qualities.** (A) Physicochemical parameters, (B) bacterial abundance and chlorophyll a, (C) microbial functioning, and (D) bacterial community structure.

To complehensively characterize the impact of CSO, analysis of temporal variations was also performed (Fig. 2 and Table S6). The temporal variations of physicochemical characteristics were not significantly different between phases, although variations tended to be greater in phase 1 than in phases 2 (PERMDISP, $P = 0.008$) (Fig. 2A and Table S6). Therefore, the effect of CSO on physicochemical water quality is similar to the effect of urban stormwater runoff. Urban stormwater runoff also collects pollutants on impervious surfaces and flows into catchment rivers (*Gobel, Dierkes & Coldewey, 2007*). *Shinya et al. (2003)* showed that runoff from elevated expressways can be a source of pollutants (*e.g.*, suspended solids, iron, and phosphorus) into the H-waterway. However, the discharge of phosphorus was not detected in phase 3 at station C (Table S9).

The temporal variations of bacterial abundance and chlorophyll a also showed no difference between phases (PERMDISP, $P = 0.009$ and 0.18, respectively) (Fig. 2B and Table S6). A previous study showed that CSO and stormwater contribute to the concentration of coliform bacteria in rivers (*Ham, Kobori & Takasago, 2009*). However,

the contributions to bacterial abundance and chlorophyll a were not confirmed in this study.

The temporal variations of microbial function were significantly greater in phase 1 than in phases 2 and 3 ($P = 0.003$ and $0.001$, respectively) (Fig. 2C and Table S6). Stormwater in phase 3 was not associated with the variation in bacterial community structure at station C.

The temporal variations in bacterial community structure were also significantly greater in phase 1 than in phase 3 ($P = 0.001$); however, these variations were not significantly different between phases 1 and 2 ($P = 0.175$) or between phases 2 and 3 ($P = 0.165$) (Fig. 2D and Table S6). The temporal variation in microbial function reflects the effect of CSOs on waterways; however, these functional changes may not be related to the changes of bacterial community structure (in "Variation in bacterial community structure"). *Roberto, Van Gray & Leff (2018)* reported that urban drainage has a marked impact on shaping benthic bacterial communities, but those changes do not affect bacterial community function. The differences between bacterial community structure shifts and bacterial functional shifts were also observed in our study.

## Variation in bacterial community structure

The heat map of relative bacterial community abundance at station C shows that *Chryseobacterium*, *Brochothrix*, *Carnobacterium*, and *Paenisporosarcina* were predominant in phase 1, while *Sediminibacterium*, *Fluviicola*, *Limnohabitans*, *Luteolibacter*, *Crenothrix*, *Rhodoferax*, and *Sulfuricurvum* were predominant in phases 2 and 3 (Fig. 3). Therefore, the bacterial community structure at station C is dynamic. In phase 1, the relative abundance of dominant genera (*e.g.*, *Chryseobacterium*, *Brochothrix*, and *Carnobacterium*) markedly declined. However, the abundances of *Sediminibacterium*, *Fluviicola*, *Limnohabitans*, and *Sulfuricurvum* were increased by CSO discharge. *Sulfuricurvum* is a putative facultatively anaerobic and sulfur-oxidizing genus, while *Sediminibacterium*, *Fluviicola*, and *Limnohabitans* are aerobic freshwater bacteria (*Kasalicky et al., 2010*; *Kodama & Watanabe, 2004*; *O'Sullivan et al., 2005*; *Ou & Yuan, 2008*). Therefore, the bacterial community composition in CSO is distinct from the bacterial community composition in the sewer; the sewer community is dominated by *Bacteroides*, *Arcobacter*, and *Acinetobacter* (*McLellan, Fisher & Newton, 2015*; *Newton & McClary, 2019*). To reduce CSO discharge after periods of prolonged heavy rain, the combined sewer structures are connected to release outlets for the H-waterway (Fig. S2). Although the size of combined sewer structures varies, the largest structure had a weir measuring 3,900 mm long × 550 mm high (*Yoshida, 2007*). Stagnant water, *i.e.*, the sewage separated by a weir after heavy rain, can promote bacterial community development during dry weather. The estimated maximum stagnant water volume in the largest structure was 0.89 m$^3$. Stagnant water discharge may explain the sudden bacterial community replacement in the waterway after the period of heavy rain (Fig. 3). Because the stagnant water volume is limited, the bacterial community shift in phase 1 at station C was temporal. A *Chryseobacterium*, *Brochothrix*, and *Carnobacterium*-dominated community was restored within 2 h. This may be explained by the dilution effect of CSOs and the subsequent stormwater runoff. The restored community had high relative abundances of

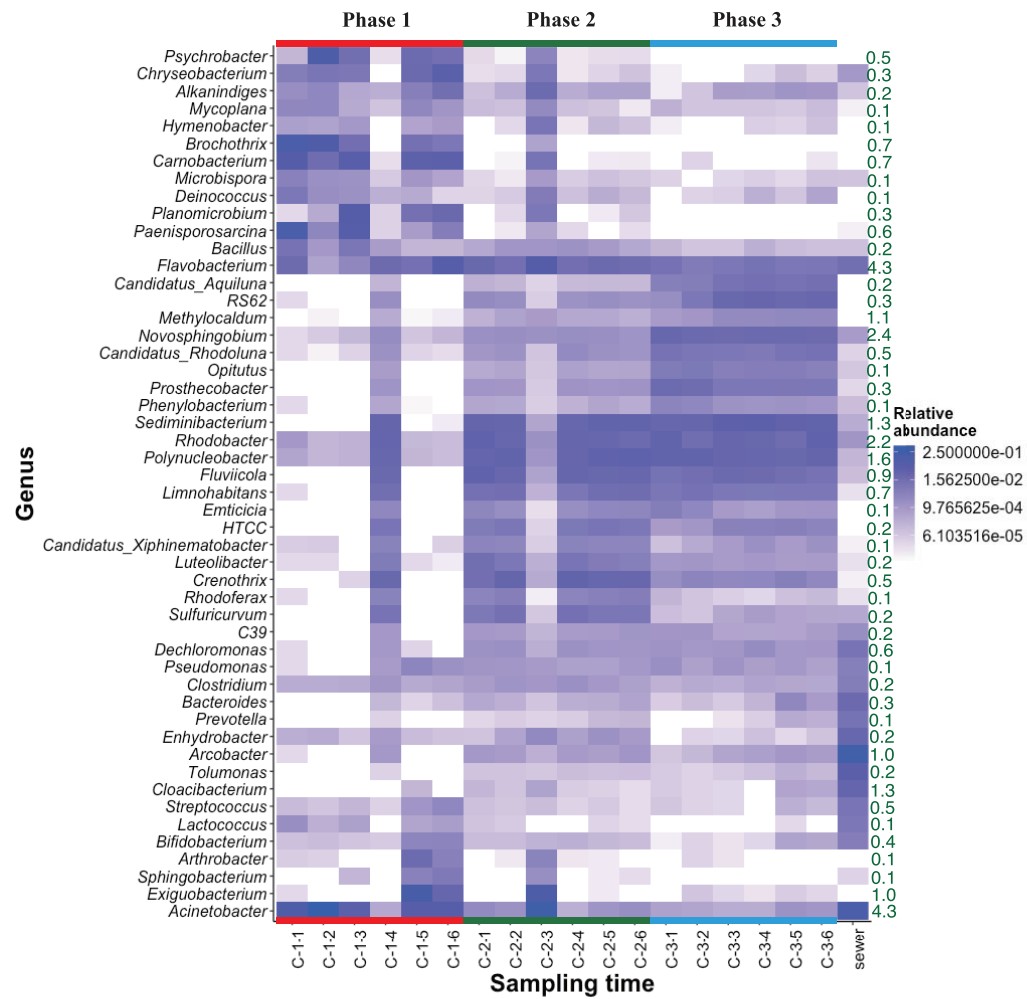

**Figure 3 Heatmap of abundant (top 50 maximum relative abundance) OTUs at station C among different phases.** The percentages of genera/phylum are represented. The detailed sampling time is shown in Table S2.

*Pseudomonas* and *Exiguobacterium* (Fig. 3). Discharged stormwater may have a high abundance of *Exiguobacterium*, which is distributed in soil and aquatic environments (*Lee et al., 2016*; *Rodrigues et al., 2009*). Bacterial community analysis in the stagnant water of combined sewer structures may shed light on the nature of the sudden bacterial community shift in phase 1.

Heat map analysis showed transient changes in bacterial abundance in phase 2 (Fig. 3). Unlike phase 1, the difference was caused by a sudden increment in the relative abundances of several genera (*e.g.*, *Exiguobacterium* and *Acinetobacter*). CSO discharge was not expected in phase 2 because the maximum hourly rainfall intensity did not exceed 3.5 mm (Table S1). The source of *Exiguobacterium* and *Acinetobacter* in phase 2 was unclear, although these genera are ubiquitous in soil. The elevated highway discharge may have affected the waterway bacterial community composition. However, this effect was presumably weak because there were no changes in bacterial community composition in phase 3, after implementation of the stormwater storage pipe. *Wyckoff et al. (2017)*

reported that roadway runoff did not influence stream water microbiological quality, while *Baral et al. (2018)* reported that ~70–75% of the microbes in storm drain outfall water came from materials washed off from street surfaces in the watershed. Thus, the impact of stormwater would presumably depend on stormwater quality. Our study site is a highly urbanized area and no discharge from agricultural fields is expected. The absence of bacterial composition changes in phase 3 indicates that storage pipe implementation suppressed discharge from the release outlet into the waterway during heavy-rainfall periods.

## Water quality in the waterway and the upper reaches of Okawa River

The water quality of the H- and D-waterways is controlled by Okawa River water inflow from the upstream gate and outflow from the downstream gate (*Nakatani, Nishida & Inoue, 2020*). The mean daily gate operating time is <2 h. This creates a semi-closed stagnant aquatic environment.

Physicochemical water quality parameters showed significant differences between stations A and C (PERMANOVA, $P = 0.001$) and stations B and C (PERMANOVA, $P = 0.002$) (Fig. S3A and Table S7). Hierarchical clustering analysis showed that the difference in physicochemical water quality was more prominent between sampling stations than between phases (Fig. 4A). Stations A and C formed independent clusters. Station B was in the station A cluster, but Station B formed a sub-cluster within the Station A-B cluster. Therefore, the physicochemical water quality parameters gradually change from upstream to downstream. A water gate in place between stations A and B may differentiate the water quality. However, continuous time series observations revealed the dissolved oxygen increase due to the intake of river water near station B (*Nakatani, Nishida & Inoue, 2020*). Thus, gate operation does not fully differentiate the physicochemical water quality between the river and the waterway.

By contrast, the physicochemical water quality is distinguished within the waterway. Water flow from stations B to C (2 km) is completed in approximately 4 days because the total length of the H- and D-waterways is 5.5 km and water exchange in the H- and D-waterways requires a mean of 11 days (*Nakatani, Nishida & Inoue, 2020*).

Because the elevated highway blocks sunlight (Fig. 1B), we hypothesized that the number of photosynthetic microorganisms differs among stations. Permutation tests of the chlorophyll *a* level and bacterial numbers showed a significant difference within the waterways (between stations B and C; PERMANOVA $P = 0.002$) (Fig. S3B and Table S7). The stagnant flow and obstruction of sunlight may affect microbial photosynthesis at station C. By contrast, the chlorophyll *a* level and bacterial abundance did not significantly differ between stations A and B (Table S7). The intake of river water may contribute to the bacterial abundance and chlorophyll *a* at station B. A recent study suggested that urbanization reduces resource use efficiency of the phytoplankton community by altering the environment and decreasing biodiversity (*Yang et al., 2022*). Our observations also suggest the altered microbial abundance within an urbanized waterway.

Hierarchical clustering analysis of bacterial abundance and chlorophyll *a* showed that the phase 2 samples at stations A and B formed an independent cluster (Fig. 4B). Since

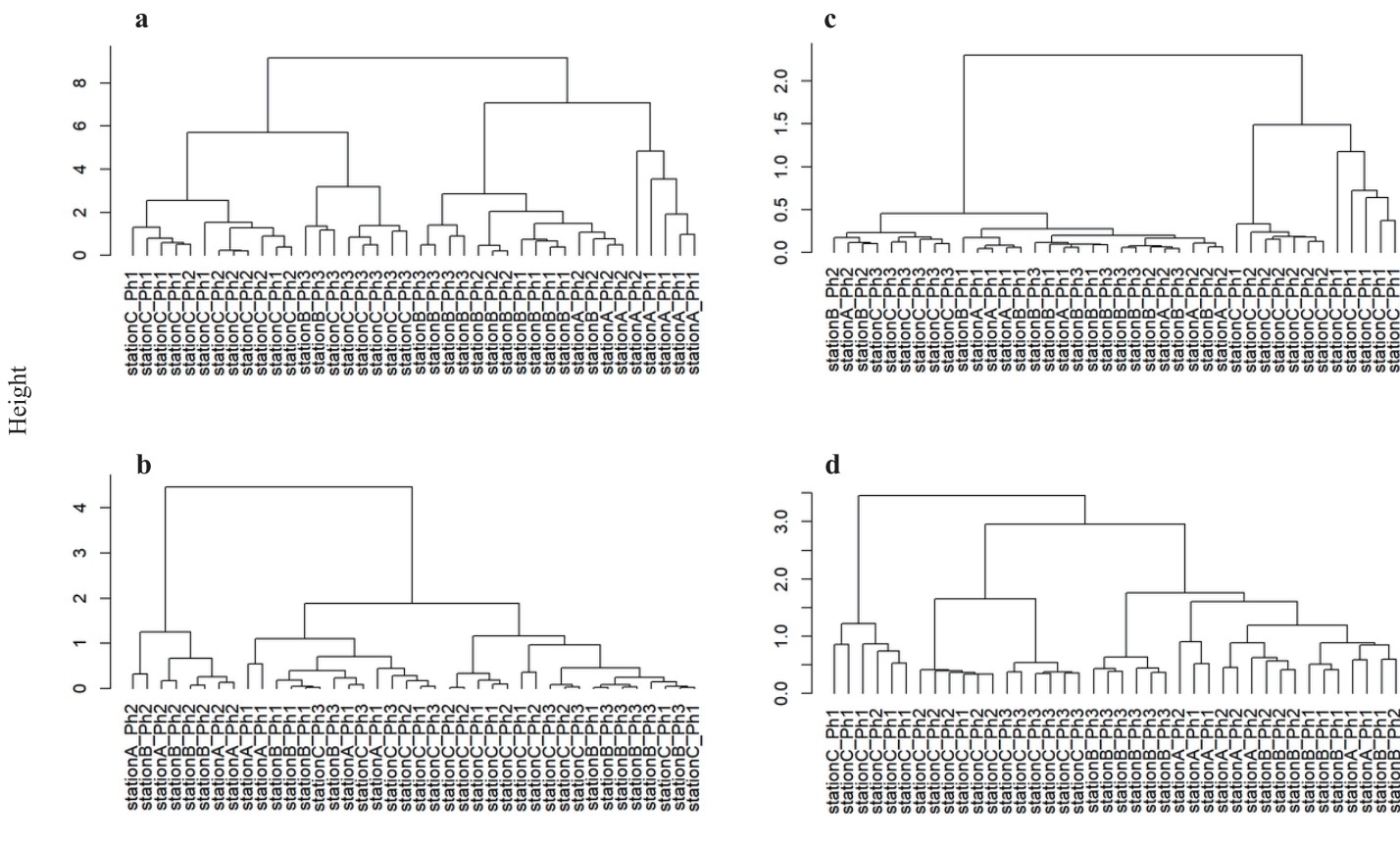

**Figure 4 Hierarchical clustering between each sample at the CSO sensitive station C.** (A) Physicochemical parameters, (B) bacterial abundance and chlorophyll a, (C) microbial functioning, and (D) bacterial community structure.

phase 2 was categorized as weak precipitation (Table S1), the chlorophyll *a* level and bacterial abundance may be sensitive to rainfall intensity at stations A and B. However, the chlorophyll *a* level and bacterial abundance at station C were not distinguished by rainfall intensity. Thus, the CSO and storm water discharge have less influence on the physicochemical water quality, chlorophyll *a* level, and bacterial abundance at station C.

Hierarchical clustering revealed that microbial community function at stations A and B was in the same cluster (Fig. 4C). Also, consumption tests did not discriminate microbial community function between stations A and B (Fig. S3C and Table S7). Unlike physicochemical water quality parameters, no gradual change was observed in microbial community function from stations A to B. Microbial community function at station C formed an independent cluster before storage pipe implementation. However, microbial community function at station C formed a cluster with stations A and B after storage pipe implementation. This indicates that CSO had a significant impact on microbial community function at station C. However, storage pipe implementation restored microbial community function, such that it was similar to upstream river water.

Hierarchical clustering analysis revealed that the bacterial community structure at station C formed two independent clusters: one composed of phase 1 data and the other

composed of the data from phases 2 and 3 (Fig. 4D). The data of phase 2 and 3 formed a clusterwith those from stations A and B, while the phase 2 and 3 data formed a sub-cluster within that cluster. Thus, the bacterial community structure at station C is different from those at stations A and B. Within the bacterial community structure at station C, the phase 1 community structure is clearly distinguishable from those in phases 2 and 3, indicating that CSO discharge had a significant impact not only on microbial community function, but also on microbial community structure. However, strage pipe implementation did not restore the microbial community structure, although pipe implementation did restore its function. Although stations A and B were in the same cluster, bacterial community structure differed significantly between them (PERMANOVA, $P = 0.002$), as well as between stations B and C (PERMANOVA, $P = 0.001$) (Fig. S3D and Table S7), suggesting that bacterial community structure also gradually changes from upstream to downstream, which was also reflected in the physicochemical water quality parameters.

The physicochemical water parameters, bacterial community composition, and microbial community function were distinguishable by sampling station, rather than phase (*i.e.*, rain intensity). However, each parameter was distinguishable among phases at station C. Therefore, CSO influenced waterway water quality; the magnitude of this change was lessened by storage pipe implementation.

### *tet* genes

Tetracycline and sulfonamide resistance genes are the most frequently detected ARGs in aquatic environments (*Amarasiri, Sano & Suzuki, 2020*). Culture-based screening revealed viable tetracycline-resistant enterobacteria in the waterway and PCR detected *tet*(A) (seven colonies), *tet*(B) (one colony), and *tet*(M) (one colony) (Table S8). The 16S rDNA sequence in one of nine colonies was >99% identical to the corresponding region in *Klebsiella pneumoniae*; the sequences in eight of nine colonies exhibited >99% identity to the corresponding regions in *Escherichia fergusonii* and related bacterial taxa.
The sequenced partial *tet*(A), *tet*(B), and *tet*(M) genes matched previously identified *tet* sequences in a database. *tet*(G), *tet*(L), *tet*(O), *tet*(Q), and *tet*(W) were not detected. Recently, antimicrobial-resistant strains of *E. fergusonii*—an opportunistic zoonotic pathogen—have been found in food animals in China and suggested as an important reservoir of antimicrobial resistance (*Tang et al., 2022*). The resistance phenotypes of sulfafurazole (97.74%) and tetracycline (94.74%) were the most frequent in the study. The isolation of tetracycline-resistant strains from the urban waterway suggests the importance of *E. fergusonii* for dissemination of antimicrobial resistance between poultry and the aquatic environment.

qPCR analysis was used to quantify *tet*(A), *tet*(B), and *tet*(M) abundances at all stations. We quantified the abundances of only three *tet* genes; thus, the results cannot be extrapolated to other antibiotic resistance systems. Higher relative abundances of antibiotic efflux pump genes (*e.g.*, *tet*(A) and *tet*(B)) than ribosomal protection genes (*e.g.*, *tet*(M)) have been reported in freshwater environments (*Yang et al., 2017*; *Yan et al., 2018*). The H-waterway (stations B and C) exhibited higher abundances of *tet*(A) and *tet*(B) (Fig. 5). Because raw sewage often contains high levels of ARGs (*Rodriguez-Mozaz et al.,*

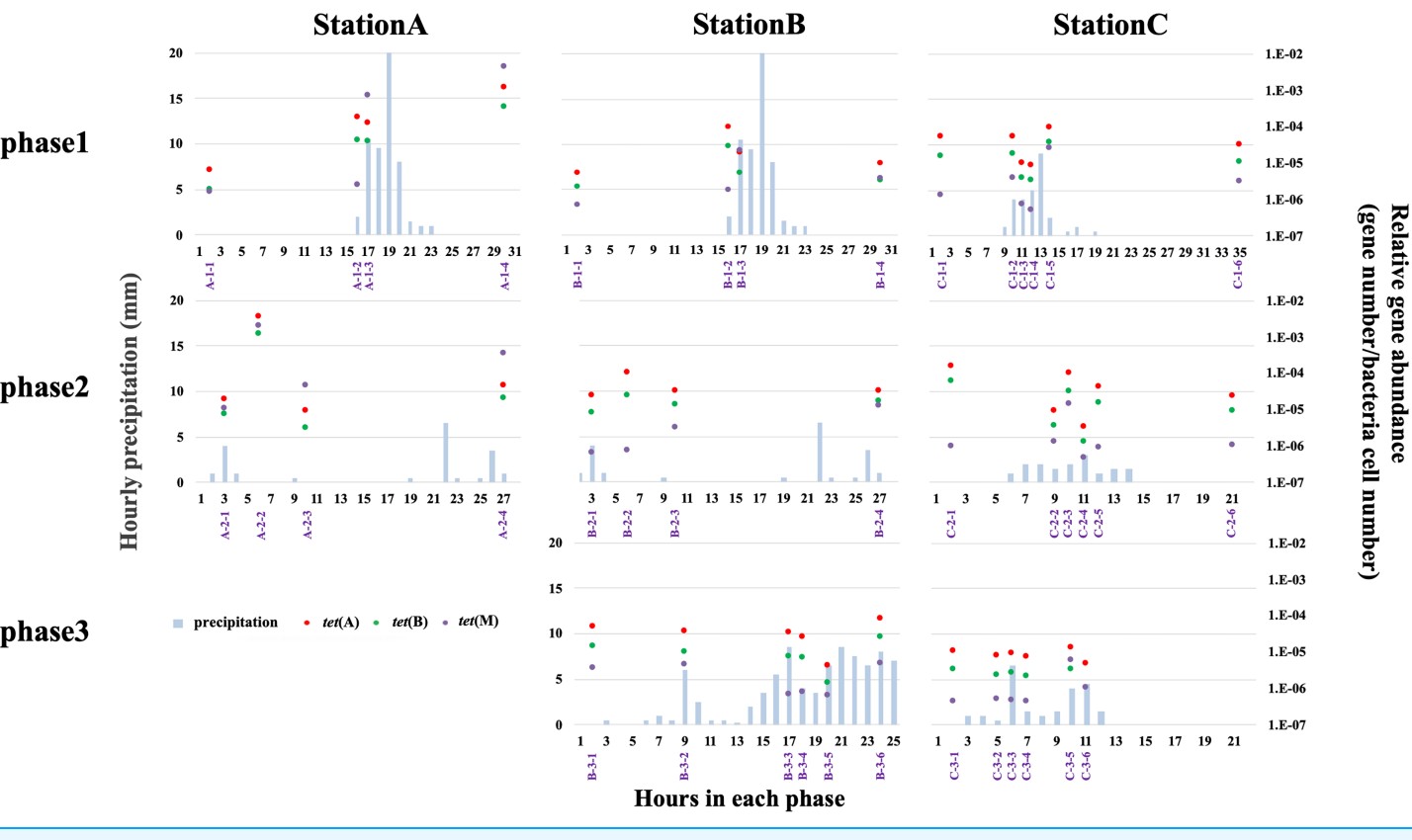

**Figure 5 Relative abundance of *tet*(A) (red-filled dots), *tet*(B) (green-filled dots), and *tet*(M) (purple-filled dots), and hourly precipitation (light-blue bar) at three sampling stations.** Timing of grab water samplings is represented with purple and shown in Table S2.

*2015*), the increase of ARGs in H-waterway is expected after the release of CSOs, which are a reported source of ARGs in the environment (*McLellan et al., 2007*; *Eramo, Delos Reyes & Fahrenfeld, 2017*). Our data from phase 1 at station C showed the decreasing relative abundance of *tet* genes until the time of the CSO (C-1-4) (Fig. 5), while the microbial community structure changed at C-1-4 (Fig. 3). The increased relative abundance of *tet* genes, especially *tet*(M), was observed at C-1-5 (Fig. 5). The timing of the increased relative *tet* abundance did not coincide with the microbial community structure changes. Phase 2 at station C showed an increased relative abundance of *tet* genes at C-2-3. Thus, the timing of the increased relative abundance of *tet* genes coincided with the microbial community structure changes in phase 2. Inter-storm variability in the timing and concentration of ARGs in CSO has also been observed (*Eramo, Delos Reyes & Fahrenfeld, 2017*). Similar to a previous study, no common patterns were observed in this study. However, compared to the concentration of *tet* genes in grabbed sewer water, the *tet* genes in the H-waterway (stations B and C) represent <10% of the *tet*(A), <4% of the *tet*(B), and <0.1% of the *tet*(M) (Fig. S4). Thus, the increased relative abundance of *tet*(M) at C-1-5 and C-2-3 may suggest sewer contamination into the waterway.

The absence of variability in the relative abundance of *tet* genes in phase 3 at station C is indicative of the storage pipe implementation during rainfall periods (Fig. 5). The data

from phase 3 in stations B and C also showed that stormwater did not contribute to the increase of *tet* genes in the H-waterway. Storm-driven transport of ARGs comprises a considerable fraction of overall downstream loadings (*Garner et al., 2017*) but this may not be the case in the H-waterway. Since many soil-dwelling bacteria are naturally antibiotic resistant (*Zhu et al., 2019*), these soil bacteria could be a source of ARGs by storm-driven transport. However, the H-waterway is a covered concrete revetment with vertical steel retaining walls. There are no soil embankments or vegetation. Moreover, the waterway flows through urban impervious surface areas and there are no agricultural fields or parks around. This fully paved environment may explain the low possibility of soil erosion and *tet* genes in the H-waterway. The variance in relative abundances of *tet* genes was more extensive at station A than at stations B and C (Fig. 5). The data from phase 1 suggest that *tet* genes were discharged from the river catchment area into Okawa River. The relative abundance of *tet*(M) exceeded the relative abundances of *tet*(A) and *tet*(B) at station A after rainfall (Fig. 5). *tet*(M) is distributed in various environments, including livestock farm soil (*Li et al., 2015*). The increased abundance of *tet* genes, particularly *tet*(M), at station A suggests that non-point discharge into a river has a marked effect on *tet* dynamics. The Okawa River is a diversion of the Yodo River, which flows through suburbs of Kyoto and Osaka. The Yodo River is soil-embanked and flows through agricultural and swine farming areas. The treatment facilities of swine urine were a major source of veterinary drugs in the Yodo River (*Hanamoto et al., 2018*), suggesting the possible discharge of ARGs from those facilities. In addition, station A is located at the confluence of the Okawa and Neyagawa rivers (see "Study site") and may be sensitive to degraded Neyagawa River water. Thus, the Okawa River (station A) shows different characteristics from the H-waterway (station B and C) in terms of *tet* gene abundance. As discussed in previous sections, physicochemical properties and microbial functions were distinguished between station C and stations A and B (Table S7). Thus, the dynamics of *tet* genes may be independent from the dynamics of physicochemical and microbial functions in the H-waterway.

## CONCLUSIONS

Our study revealed the impact of CSO on an urban waterway. CSO has significant impacts on microbial function and bacterial community structure in the waterway, while the contribution of CSO to physicochemical parameters, bacterial abundance, chlorophyll a were not confirmed. The impact of CSO on the waterway bacterial community structure was temporal and the bacterial community composition in CSO was distinct from that in sewers. Changes in the relative abundance of *tet* genes, especially *tet*(M), were observed after CSOs, but did not coincide with changes in the microbial community composition, suggesting that the parameters affecting the microbial community composition and the relative abundance of *tet* genes differ. The physicochemical water quality, bacterial community composition, and microbial community function in the waterway were also distinguishable from the upper reach of the river. After pipe implementation, however, the impact of CSO on the bacterial community structure, bacterial community composition,

and the abundance of *tet* genes in the waterway was ameliorated. These findings support the efficiency of storage pipe operation in waterway management in urban areas.

## ACKNOWLEDGEMENTS

We thank H. Kawamura, Y. Yokoyama, K. Kamei, J. Yamanaka, Y. Nakaguchi, M. Eguchi, and A. Taniguchi for laboratory work and analytical measurements. We also thank K. Takehara, T. Fumoto, and the staff of the Laboratory of Urban Microbial Ecology for assistance with field and laboratory work. Finally, we thank the Osaka Public Works Bureau for assistance with field experiments.

### Funding

This study was supported by the Japan Sewage Works Association Foundation. Kazuaki Matsui was supported by the River Fund of The River Foundation, Japan and by JSPS KAKENHI Grant No. 20H04348. Takeshi Miki was supported from Sumitomo Foundation for the research grant entitled "Functional evaluation of whole ecological communities using time series data for effective lake and river conservation". The funders had no role in study design, data collection and analysis, decision to publish, or preparation of the manuscript.

### Grant Disclosures

The following grant information was disclosed by the authors:
Japan Sewage Works Association Foundation.
River Fund of The River Foundation, Japan.
JSPS KAKENHI: 20H04348.
Sumitomo Foundation.

### Competing Interests

The authors declare that they have no competing interests.

### Author Contributions

- Kazuaki Matsui conceived and designed the experiments, performed the experiments, analyzed the data, prepared figures and/or tables, authored or reviewed drafts of the article, and approved the final draft.
- Takeshi Miki analyzed the data, prepared figures and/or tables, authored or reviewed drafts of the article, and approved the final draft.

### DNA Deposition

The following information was supplied regarding the deposition of DNA sequences:

The raw FASTQ files are available at the DDBJ DRA BioProject: PRJDB10469.

The 16S rDNA and tetracycline-resistance gene sequences are available at the DDBJ: LC611439 to LC611447 and LC610784 to LC610786.

## Data Availability

The raw data are available in the Supplemental Files.

## Supplemental Information

Supplemental information for this article can be found online at http://dx.doi.org/10.7717/peerj.14684#supplemental-information.

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
