# Peer review of "Microbial community composition and function in an urban waterway with combined sewer overflows before and after implementation of a stormwater storage pipe"

_PeerJ, doi:10.7717/peerj.14684_

## Round 0.1 · original submission · Major Revisions

Dear Drs. Matsui and Miki:

Thanks for submitting your manuscript to PeerJ. I have now received two independent reviews of your work, and as you will see, the reviewers raised some concerns about the research. Despite this, these reviewers are optimistic about your work and the potential impact it will lend to research on waterway microbial communities. Thus, I encourage you to revise your manuscript, accordingly, taking into account all of the concerns raised by the reviewers.

Please improve the content and clarity of the figures and tables (see suggestions by the reviewers). Please also ensure that all appropriate references are included. Overall, the presentation of the manuscript needs improvement, but the authors have provided plenty of suggestions that I feel will greatly improve your manuscript.

Please note that reviewer 2 has included a marked-up version of your manuscript.

I look forward to seeing your revision, and thanks again for submitting your work to PeerJ.

Good luck with your revision,

-joe

Reviewer 1 ·

Basic reporting

The language used, especially in the construction of the sentences and the vocabulary used in the Results and Discussion sector are not exactly correct and comprehensible. Some examples where the language could be improved include lines 308-309, 327-328, and 384-386; the current phrasing together with the lack of accuracy (as explained below) of the figures makes comprehension of the result difficult.

Experimental design

no comment

Validity of the findings

It is not clear how some of the analyses can help in answering the scientific questions.

Additional comments

Despite the fact that the experimental design investigate interesting topics of considerable scientific relevance for the journal audience, the language used, especially in the construction of the sentences and the vocabulary used in the Results and Discussion sector are not exactly correct and comprehensible. Some examples where the language could be improved include lines 308-309, 327-328, 384-386; the current phrasing together with the lack of accuracy (as explained below) of the figures makes comprehension of the result difficult.
Moreover in the Result and Discussion paragraph is not clearly explained the use and the results obtained with the data concerning Physicochemical parameters, Bacterial abundances and chlorophyll a, microbial functioning and community structure. The statistical analysis and the figure generated don’t explain how the author use the large amount of data available: for example the Figure 2, is not clear how the author obtain the result, and how the plot are made.
Furthermore, the physico-chemical parameters have been expressed as totals, and are not reported as individual values in the manuscript, The suggestionis to specify how they are used to arrive at the total and to insert them at least in the Suppl. Information.
The microbial functioning are not well described in “Materials and Methods” and more important they are not well described and discussed in the dedicated paragraph. I suggest you to improve the description at lines 413- 427 to provide more justification for your analysis and results.
On the contrary the paragraphs describing the tetracycline-resistant bacteria were thoroughly described, with a good focus on the discussion and the scientific implications resulting of the finding results. At the same level, the Figure 5 and the Supplementary figures concerning the Tetracycline analysis appear easy to understand and accurate.

Fig. 2
The legend is missing, and it is not clear what “distance” in the X axes means.

Fig. 3
Is not clear what the term genera/phylum mean?

Fig. 4
The figures include all the sampling sites and not only the station C

Fig. S2
Analysis not described and used in the manuscript

·

Basic reporting

MS is clear and professional English, the literature is well referenced and relevant and the structure conforms to PeerJ standards. The research question is defined and relevant but many unclear sentences and very generic

Experimental design

Method described with sufficient detail and information.

Validity of the findings

In the section Results and Discussion, the MS is a list of data little discussed (even if all data have been provided and they are robust, statistically sound), and an approximate final conclusion. It seems more a Report than a scientific manuscript, despite the interest in the topic, a good methodological approach, and excellent results. All the text has many unclear sentences and very generic.
I suggest that the MS needs a rewriting in some sentences in the discussion and also a new conclusion, because are not well stated,

Additional comments

no additional comments

---

## Round 0.2 · accepted · Accept

Dear Drs. Matsui and Miki:

Thanks for revising your manuscript based on the concerns raised by the reviewers. I now believe that your manuscript is suitable for publication. Congratulations! I look forward to seeing this work in print, and I anticipate it being an important resource for researchers studying waterway microbial communities. Thanks again for choosing PeerJ to publish such important work.

Best,

-joe

Reviewer 1 ·

Basic reporting

The authors have excellently corrected and re-presented the manuscript using language that is appropriate and understandable, and the experimental design and the achieved result are properly expressed and described with well-referenced and relevant literature.
The structure of the manuscript conforms to PeerJ standards.
The proposed corrections and suggestions were accepted and adequately resolved, both in the main text as well in the figures.

Experimental design

As mentioned in the first evaluation the experimental design investigates interesting topics of considerable scientific relevance for the journal audience.
The adjustments in the description of the methodologies together with the changes made in the figures description and the additional material provided facilitate the understanding of the text and the analyses performed.

Validity of the findings

Unfortunately, the conclusions expressed, as also specified in the final paragraph, did not always clear all the doubts. but the usefulness of the analysis together with the results achieved in the part concerning the Tetracycline and sulphonamide resistance genes makes it possible, with a few minor adjustments and a greater focus on it to obtain a valid and acceptable article.